# A Weighted Stochastic Conjugate Direction Algorithm for Quantitative Magnetic Resonance Images—A Pattern in Ruptured Achilles Tendon T2-Mapping Assessment

**DOI:** 10.3390/healthcare10050784

**Published:** 2022-04-23

**Authors:** Piotr A. Regulski, Jakub Zielinski, Bartosz Borucki, Krzysztof Nowinski

**Affiliations:** 1Department of Dental and Maxillofacial Radiology, Faculty of Medicine and Dentistry, Medical University of Warsaw, 02-091 Warsaw, Poland; 2Interdisciplinary Centre for Mathematical and Computational Modelling, University of Warsaw, 00-927 Warsaw, Poland; jziel@icm.edu.pl (J.Z.); babor@icm.edu.pl (B.B.); know@icm.edu.pl (K.N.)

**Keywords:** quantitative T2-map, biexponential method, weighted reconstruction, Achilles tendon rupture, MRI

## Abstract

This study presents an accurate biexponential weighted stochastic conjugate direction (WSCD) method for the quantitative T2-mapping reconstruction of magnetic resonance images (MRIs), and this approach was compared with the non-negative-least-squares Gauss–Newton (GN) numerical optimization method in terms of accuracy and goodness of fit of the reconstructed images from simulated data and ruptured Achilles tendon (AT) MRIs. Reconstructions with WSCD and GN were obtained from data simulating the signal intensity from biexponential decay and from 58 MR studies of postrupture, surgically repaired ATs. Both methods were assessed in terms of accuracy (closeness of the means of calculated and true simulated T2 values) and goodness of fit (magnitude of mean squared error (MSE)). The lack of significant deviation in correct T2 values for the WSCD method was demonstrated for SNR ≥ 20 and for GN–SNR ≥ 380. The MSEs for WSCD and GN were 287.52 ± 224.11 and 2553.91 ± 1932.31, respectively. The WSCD reconstruction method was better than the GN method in terms of accuracy and goodness of fit.

## 1. Introduction

Quantitative T2 mapping is used to measure the transverse relaxation time T2 and assess degenerative and reparative processes in musculoskeletal tissues such as tendons, ligaments, articular cartilage, and muscles [1,2]. This technique is used to assess Achilles tendon (AT) ruptures, cruciate ligament tears, lumbar disc degeneration, and even to detect myocardial infarction [3,4,5,6,7]. Quantitative T2 mapping has attracted much attention in recent years due to the incorporation of corresponding sequences into generally available MRI scanners (e.g., the CartiGram sequence—GE and the MyoMaps sequence—Siemens) [8,9].

A T2 map consists of relaxation times calculated for every image voxel. The standard approach for obtaining T2 times requires acquiring at least two different echo time (TE) images and fitting them to a given minimization model. The most commonly used fitting methods are based on non-negative, nonlinear least squares regression with a monoexponential function. Currently, the Gauss–Newton (GN) numerical optimization method is the most frequently used in radiological workstations. However, this method significantly increases the noise level of the reconstructed images, which results in a reduction in the reconstruction accuracy [10,11]. This is of particular importance for low signal-to-noise ratio (low SNR) MRIs [12].

Contrary to the commonly used monoexponential reconstruction method, we present a new method of calculating so-called biexponential T2 maps. Our method, a weighted stochastic conjugate direction method (WSCD), can also be successively used for monoexponential reconstruction; however, its use as a biexponential method has many more benefits. This approach enables the differentiation of two T2 components (short and long components) for a single type of biological tissue. The difference between these two components reflects the local anisotropy and subvoxel inhomogeneity of the tissue, especially during its regeneration. The most challenging issue related to the biexponential approach is dealing with the effects of noise on the stability of the reconstruction results. Another issue is associated with the elongation of the examination and reconstruction times compared to those in the monoexponential method. Our method addresses both issues. We propose innovative concepts such as the introduction of weights into a biexponential model to reduce noise and increase the accuracy of our method and the use of a novel stochastic method to shorten the reconstruction time.

To verify the outstanding accuracy and quality of the new approach, it was compared to the standard GN method based on synthetic data simulations and MRIs of the postrupture AT. The GN approach can also be used in biexponential reconstruction; however, both the noisiness and reconstruction time are increased [13]. T2 maps of ATs of high clinical importance are used in the assessment of the healing process [13,14]. The biomechanical and biochemical changes in the collagen fibers of the tendon that occur during rupture and recovery lead to significant changes in T2 time. An operative, surgical approach for repairing postrupture AT, which has been gaining popularity, requires MRI for differential diagnosis, surgical treatment planning, and regeneration monitoring follow-up [15]. Therefore, postrupture AT MRIs appear to be an ideal basis for quality assessments of different T2-mapping approaches, especially more sophisticated approaches [16].

The incidence of AT rupture is high—approximately 7–18 per 100,000 in the general population—and it may occur during spontaneous recreational activity or during professional sports activities [17,18]. Compared to that for other tendons and ligaments, the AT healing process is prolonged due to comparatively poor blood supply [19,20]. Major changes in the AT mainly occur during the first half year after injury, resulting in changes in the MRI of the tendon [21].

Therefore, the aim of this study was to present a new WSCD method and assess its accuracy based on simulated data and AT rupture MRIs.

## 2. Materials and Methods

### 2.1. Calculation of T2 Time

A biexponential signal can be considered a sum of two monoexponential signals containing the short T2 relaxation time (*T_S_*), long T2 relaxation time (*T_L_*), short signal amplitudes (*A_S_*), and long (*A_L_*) signal amplitudes. The signal is affected by noise (*η(t)*). Therefore, the relationship between the above-mentioned parameters and the signal function (*f*) for a given echo time (*t*) is defined as [11]:(1)$$f\left(t\right)=A_{S}\exp\left(-\frac{t}{T_{S}}\right)+A_{L}\exp\left(-\frac{t}{T_{L}}\right)+\eta \left(t\right)$$

We assumed that the noise has an impact on the input signals to different degrees. To compensate for the differences in noise levels among input signals, weights were introduced in our WSCD method. The model function, considered for a single point *P* in an image, is given by:(2)$$f_{P}\left(t\right)=W_{P}\left(t\right)\left[A_{S_{P}}\exp\left(-\frac{t}{T_{S_{P}}}\right)+A_{L_{P}}\exp\left(-\frac{t}{T_{L_{P}}}\right)+\eta_{P}\left(t\right)\right]$$
where *t* is the *TE*; *T_SP_* and *T_LP_* are the *T*_2_ relaxation times of the short- and long-time components, respectively; *A_SP_* and *A_LP_* are the signal amplitudes of the short- and long-term components, respectively; *η(t)* is noise; and *W(t)* is a weight [22]. The last term in the model function is a multiplication term related to noise, which is assumed to be homogenous, and weight factors. The average noise level (*n*) is therefore constant.

The weights are calculated within a window of radius r. The window is defined as a neighborhood U of point P with dimensions of (2 × r + 1)^2^ pixels. It is assumed that point Q belongs to the neighborhood U and is not equal to the point P (Figure 1).

The weights are calculated in *U* for point *P* and are defined by:(3)$$W_{P}\left(t\right)=\frac{n_{TE}\cdot \sum_{Q\in U}\left[\alpha_{P,Q}\left(t\right)\omega_{P,Q}\left(t\right)\right]}{\sum_{t=1}^{m}\sum_{Q\in U}\alpha_{P,Q}\left(t\right)}$$
where *n_TE_* is the number of echo times; *α_P,Q_* is the weighting factor; and *ω_P,Q_* is the normalization factor between points *P* and *Q*. The weighting factor is used to increase the total weight if the difference between the signal intensities at points *P* and *Q* is small, which occurs in the case of low noise. Consequently, images with lower noise levels are promoted, causing an increase in the total SNR for the resulting T2 map, even if the SNRs of the input images are low. These factors are calculated with the following equations:(4)$$\alpha_{P,Q}\left(t\right)=\exp\left(-\frac{||P,{Q||}^{2}}{\sigma^{2}}\right)\cdot \exp\left(-\frac{{\left(y_{P}\left(t\right)-y_{Q}\left(t\right)\right)}^{2}}{\sigma^{2}}\right)$$
(5)$$\omega_{P,Q}\left(t\right)=\frac{y_{Q}\left(t\right)}{y_{P}\left(t\right)}$$
where *y_P_(t)* is the input signal at point *P*; *y_Q_(t)* is the input signal at point *Q*, ||*P,Q*|| is the distance between points *P* and *Q*; and σ is the standard deviation of the noise [22]. Weights were introduced to decrease the influence of signal noise. The local noise measure was assumed to be a variance (*σ*^2^) calculated for each voxel and each echo time within a window of a given radius. Therefore, to avoid overestimation of the noise within windows, the variance was used in the denominator of the above-mentioned equation. The normalization factor was used to reduce the impact of noise on the considerably different signal values at points *P* and *Q*. A similar approach was used in anisotropic denoising algorithms [23,24].

The function used for minimization with our WSCD model was the mean squared error (MSE) of the fit between the model function (*f_P_(t)*) and the given input signal (*y_P_(t)*) [25]:(6)$$g_{P}\left(\theta \right)=\overset{m}{\underset{t=1}{\sum }}{\left[f_{P}\left(t\right)-y_{P}\left(t\right)\right]}^{2}$$
where *θ* is the set of estimated parameters, which are the relaxation times of the short- and long-term components (*T_SP_* and *T_LP_*), signal amplitudes (*A_SP_* and *A_LP_*), and noise (*n_P_*). Based on Equations (2) and (6), the minimization function for the above-mentioned parameters is given by:(7)$$g\left(T_{S_{P}},T_{L_{P}},A_{S_{P}},A_{L_{P}},n_{P}\right)=\overset{m}{\underset{t=1}{\sum }}{\left[W_{P}\left(t\right)\cdot A_{S_{P}}\exp\left(-\frac{t}{T_{S_{P}}}\right)+W_{P}\left(t\right)\cdot A_{L_{P}}\exp\left(-\frac{t}{T_{L_{P}}}\right)+n_{P}-y_{P}\left(t\right)\right]}^{2}$$

The WSCD method requires a set of initial parameters and initial search vectors. The initial parameters are calculated by monoexponential log-linear regression at time *T*_2_ and for amplitude *A*. The initial *T_S_* and *T_L_* parameters are equal to 75% and 125% of monoexponential *T*_2_, respectively, and *A_S_* = *A_L_* = *A*/2. The initial noise level is set to zero. Therefore, the initial set of parameters is defined as:(8)$$B_{0}(0.75T,\;1.25T,\;A/2,\;A/2,\;0\}$$

The initial search vector values are equal to 10, and their normals are aligned to each axis. In each step, a line search is performed from the initial set of parameters (point *B*_0_) to the end point ($B_{1}^{i}$) along the search vector (*v^i^*). Any point located between *B*_0_ and $B_{1}^{i}$ can be defined as:(9)$$B_{s}^{i}=B_{0}+\vec{v^{i}}\cdot s$$
where *s* ∈ <0,1> [26].

The segment |B_0_B^i^_1_| is divided into *m* = 20 subsegments that are equal in length. Let us consider the set of points (*C^i^*) containing points *B_0_* and $B_{1}^{i}$ and the set of points $B_{k}^{i}$ for which the value of the function $g\left(B_{k/m}^{i}\right)$ is larger than the function value at two adjacent points $g\left(B_{\left(k-1\right)/m}^{i}\right)$ and $g\left(B_{\left(k+1\right)/m}^{i}\right)$; *k* ∈ (0,*m*). For each subset of the set *C^i^*, which includes two adjacent points (*C^i^_j_*, *C^i^_j+1_*), a minimum *M_j_* is found between these points during the line search with the Brent method [26]:(10)$$M_{j}\left(C_{j}^{0}+\vec{C_{j}^{0}C_{j+1}^{0}}\cdot \gamma^{0},\;\;C_{j}^{0}+\overset{1}{\underset{i=0}{\sum }}\vec{C_{j}^{i}C_{j+1}^{i}}\cdot \gamma^{i},\;\ldots ,\;C_{j}^{0}+\overset{\kappa }{\underset{i=0}{\sum }}\vec{C_{j}^{i}C_{j+1}^{i}}\cdot \gamma^{i}\right)$$
where *κ* = 4 is the number of parameters reduced by one and *γ_i_* is the scalar determined from the model function during the line search along vector $\vec{C_{j}^{i}C_{j+1}^{i}}$. The new point *D_j_* obtained from the minimum *M_j_* is:(11)$$D_{j}=C_{j}^{0}+\overset{\kappa }{\underset{i=0}{\sum }}\vec{C_{j}^{i}C_{j+1}^{i}}\cdot \gamma^{i}$$

The new displacement vector becomes a new search vector (*v^i^_j_*), and the search vector that contributes the most to the new direction is deleted. Both the new *D_j_* and *v^i^_j_* are added to the list. In the next iteration, the point at which the model function has the lowest value and corresponding vector are pulled from the list. The other nine points and vectors are randomly selected from the list. For all ten points and vectors, the new minima, points, and search vectors are calculated and added to the list. The algorithm iterates 200 times or until no significant improvement is obtained, defined as a difference between the parameters in the previous iteration and current iteration, equaling less than 0.0001. The point associated with the lowest value of the model function is considered the global minimum.

### 2.2. Simulation Data

The data used to simulate the signal intensity with biexponential decay were produced with Equation (1). Considering both the bias and the acquisition time, the simulation data were created with eight TEs: 9, 18, …, 72 ms. To simulate real conditions, we created 100 × 100 pixel intensity images that consisted of 25 subimages (with a resolution of 20 × 20 pixels). Each subimage differed in terms of short and long relaxation times and signal amplitudes. According to the method proposed by Anastasiou and Hall [27], noise with a Rician distribution was added for SNR < 7, and noise with a Gaussian distribution was added for SNR ≥ 7. Several standard deviations (SDs) of noise were chosen, namely, σ ∈ 50, 75, 150, 300, and 600, resulting in five sets of eight signal intensity images (Figure 2). The arbitrary selection of noise levels resulted in constant SNR values throughout the image. The SNR in the simulated data was defined as the quotient of the signal value and noise value. Therefore, the images had SNR values equal to 60, 40, 20, 10, and 5. The results were analyzed according to these SNR values.

The GN with the model function given in Equation (1) was used as a reference for the WSCD method, and the accuracy was assessed for both methods. Accuracy was defined as how close to the mean calculated relaxation times for the short and long time components were to the true T2 values. This metric was calculated separately for each SNR level. A statistical analysis was performed with the Wilcoxon signed-rank test (WSRT), and the 95% confidence intervals over the pseudomedian were estimated. The mean differences and SDs of *T_S_* and *T_L_* values were also calculated for each method and SNR level.

### 2.3. MRI Acquisition Protocol

Fifty-eight postrupture AT patients underwent magnetic resonance examinations one week after surgery. The MRI procedure was performed on a 1.5-T MRI unit (Sigma HDxt, GE Medical Systems, General Electric, Chicago, IL, USA) using an eight-channel phased-array transmit/receive leg coil (HD Lower Leg Coil, General Electric, Chicago, IL, USA). For the T2 time measurements, the fast multiple spin echo sequence was used. The T2 map imaging parameters were as follows: repetition time = 1200 ms, eight TEs = 9, 18, …, 72 ms, field of view = 150 × 150 mm^2^, matrix = 512 × 512 voxels, number of slices = 10, slice thickness = 3.5 mm, spacing between slices = 4.2 mm, average acquisition time = 8:17 min, and in-slice resolution = 0.29 × 0.29 mm^2^. All subjects provided written informed consent, and the study was approved by the regional research ethics board. A detailed characterization of the obtained signals is presented in Figure 3.

The AT region was segmented with a multistep segmentation algorithm based on the region growing approach described in detail in [21]. The algorithm consists of adaptive thresholding for monoexponentially reconstructed T2 maps, automatic placement of seed points, seed region growing, and morphological closing operations. Biexponential T2-map reconstructions were performed for segmented AT regions (Figure 4).

To confirm the goodness of fit of our method, an analysis of MSEs was performed separately for the WSCD and GN models based on MRI data. Each MSE was calculated as the square of the measured signal minus the fitted value. To determine whether one of the biexponential models provided a better fit, MSEs were compared, and the model with the lower MSE was selected as more suitable. The MSEs were compared between the models with the WSRT. For the accuracy assessment, the SNRs of the WSCD and GN methods were compared.

To compare the accuracy of both methods based on real MRI data, the resultant SNR was calculated using the background noise in air as a reference. The signal was the average value from the segmented AT T2 map. Noise was defined as the mean value in the background. The SNR values were compared with the WSRT results. A statistical analysis was performed with RStudio.

## 3. Results

The accuracy assessment performed based on the simulated data and subsequent WSRT showed that the results of the WSCD method for cases with a SNR greater than 20 did not show a significant deviation from the true T2 values. The detailed results are shown in Table 1. For SNRs less than 20 in the WSCD method and for SNRs less than 380 in the GN method, the WSRT indicated that the results significantly deviated from the true T2 values (*p* < 0.0001).

The GN method displayed a tendency to overestimate T2 values for all SNRs for the short time component. The lack of significant deviation in correct T2 values was proven for the GN method for SNR ≥ 380. The relations between true values estimated with the means and SDS of the WSCD and GN short and long relaxation times are shown in Figure 5. A comparison of the T2 map reconstruction results obtained with the WSCD and GN methods for simulated data is presented in Figure 6.

The analysis of both methods was performed based on 580 separate slices of AT MRIs. The mean SNR value of the initial MRIs was 26.59, and therefore, according to the simulation data, T2 maps reconstructed with the WSCD method for real MRIs of the postrupture AT region were above the lower threshold of the SNR (SNR > 20) and did not show a significant deviation from the true T2 values. T2 maps reconstructed with the GN approach were below the lower threshold of the SNR, thus significantly deviating from the true T2 values.

The averages of the *T_S_* and *T_L_* values in the AT region obtained with the WSCD method were 12.52 ± 9.67 and 70.41 ± 46.12, and those obtained with the GN method were 17.16 ± 10.54 and 75.70 ± 70.51, respectively. An example of MRI reconstruction is shown in Figure 7. The average values of noise, calculated for the air background, for *Ts* and *T_L_* obtained with the WSCD method were 0.49 ± 3.95 and 2.75 ± 7.54, and those obtained with the GN method were 5.58 ± 10.54 and 25.75 ± 24.51, respectively.

The mean value of the SNR for the resultant reconstructions was 25.51 for the WSCD method and 3.02 for the GN method, verifying the outstanding accuracy of the WSCD approach. The WSRT statistical analysis revealed a significant deviation of the GN model from the WSCD model (*p* < 0.0001). An example SNR analysis is presented in Figure 8.

The mean and SD of the MSE in the AT region for the WSCD and GN methods were 287.52 ± 224.11 and 2553.91 ± 1932.31, respectively. Notably, a lower MSE was obtained for the WSCD method, verifying that it provided the best goodness of fit. The WSRT revealed that the mean MSE of the WSCD model significantly deviated from that of the GN (*p* < 0.0001) model.

One processing step in the GN and WSCD methods for a single window required approximately 3.11 ± 0.34 ms and 0.55 ± 0.28 ms per voxel on a computer station with an 8-core Intel Xeon Processor E5-2687-W (Intel Corporation, Santa Clara, CA, USA), respectively.

## 4. Discussion

The WSCD approach is similar to the algorithm described by Powell in [28]. We introduced several improvements for quantitative MRI reconstruction. The first improvement was related to the bidirectional line search. In the original algorithm, only one minimum in a segment, defined by the initial point and the search vector, was found. Our algorithm found all local minima in a segment. Contrary to the original algorithm, we implemented a list that stored all of the positions and search vectors, which were added in each iteration. During subsequent iterations, ten positions and vectors were selected from the list. If a new position showed significant improvement, it was added to the list. The algorithm iterates until the list is empty or an arbitrarily selected iteration limit is exceeded.

The second improvement was the introduction of weights, which greatly compensate for the differences in noise levels between the input signals (obtained with different TEs), and are calculated slice by slice and processed with a convolutional model function in a given window; the result is directly proportional to the sum of differences between the signal values of the window center pixel and neighboring pixels inside the window. Weights are used to avoid overestimation of model function parameters, which is often caused by noise, especially in windows incorporating regions with considerably different signals such as between tissues with distinguishable signals.

The WSCD method presented in this study proved to be more accurate than the commonly used GN method for signals with SNRs greater than or equal to 20. This reconstruction method provides exact results for short and long T2 components, especially for postrupture, postsurgery AT images. The level of signal in the ATR region in all eight TE images were proven sufficient for performing reconstruction. The presented approach was characterized by the lowest MSE, which was significantly better than that of the GN method.

Notably, there are three possible results that can be obtained in terms of the T2 times for a single voxel in a T2 map: (I) both requested T2 times are extracted; (II) only one T2 time is extracted; or (III) some T2 times are close to infinity. Case I implies the occurrence of biexponential signal decay. Case II occurs when a monoexponential or linear approximation is obtained. Case III occurs when the approximation ends with the best fit being a constant function. For ruptured ATs, the first case was found to occur most commonly. This finding suggests that most tissues in the ruptured AT region are inhomogeneous. This inhomogeneity could be caused by the healing process of the AT when the collagen fibrils are rearranged and extracellular matrix is produced by fibroblasts [3]. The histological and macroscopic construction of AT tissue changes during its regeneration. Therefore, biexponential WSCD T2 maps reflect the complex structure of the AT with higher accuracy than monoexponential maps, and the WSCD approach might be useful for reconstruction, especially in the assessment of AT regeneration.

Two main difficulties are encountered in calculating T2 maps, as mentioned in the literature, namely, the high computational complexity and the divergence of the output T2 times, which is highly influenced by the initial parameters of the minimization model [29,30]. In this paper, we showed that the computational complexity could be decreased by employing the WSCD method because no derivatives have to be calculated. Furthermore, multiple minima recognition and assessment in the WSCD method helps avoid the need to find a local minimum instead of a global minimum.

Attempts have been made to solve the above-mentioned problems with monoexponential models. Improvements in the accuracy and precision of low-SNR T2 maps were obtained by Raya et al. [10], who developed two noise-corrected fitting methods: fitting to a noise-corrected exponential and fitting the noise-corrected squared signal intensity to an exponential. Sandino et al. [7] introduced a pixelwise nonlinear regression method by using SNR-scaled image reconstruction and truncating low-SNR measurements. Akcakaya et al. [31] developed an improved T2-based, balanced steady-state free-precession sequence and a signal relaxation curve fitting method. To the best of our knowledge, a few quality-improving methods for biexponential models have also been presented including methods based on optimized TE sampling procedures [27,32], nonselective radio frequency pulses [12], and separate nonlinear filters [33,34,35]. Huang presented a study in which a nonlinear mixed-effect model was used for the reconstruction of biexponential time maps, and it improved the accuracy of parameter estimation [36]. Shao et al. [37] used a maximum likelihood estimation algorithm for noise estimation in a biexponential approach. However, none of these methods considered a postacquisition optimization or assessment of the reconstruction model.

The clinical aspect of this research was discussed by Kapinski et al. [38], who monitored the healing of postrupture ATs. The clinical application of this method is expected to provide physicians and surgeons with faster and more effective information during the prediagnosis process. Furthermore, the presented biexponential reconstruction was used in assessments of the anatomical structure of the temporomandibular joint [39].

## 5. Conclusions

In conclusion, the method presented in this manuscript for calculating weighted biexponential T2 maps proved to be accurate for SNR ≥ 20, showed the best goodness of fit, and displayed a shorter computational time than other methods. The WSCD method incorporates noise reduction based on suitable calculated weights, and the desired noise–blur balance can be achieved by choosing an adequate window size. Therefore, it is possible to adapt the parameters of the acquired image according to the requirements of the application such as a quantitative assessment of a particular tissue, a visual assessment by a radiologist, the segmentation of selected organs, or further analysis based on convolutional neural networks. Furthermore, our approach makes it possible to incorporate the WSCD reconstruction method into qualitative healing assessments of ruptured Achilles tendons.

## Figures and Tables

**Figure 1 healthcare-10-00784-f001:**
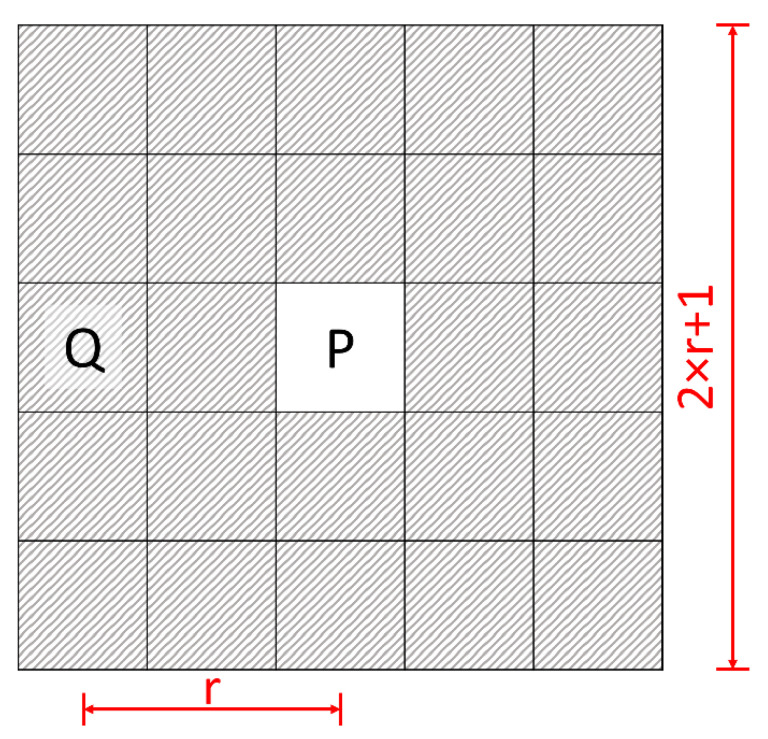
Schematic diagram of the neighborhood of point *P*. Neighborhood *U* is marked with stripes. Point *Q* is in the neighborhood *U* of *P*. The weights are calculated within a window of radius r.

**Figure 2 healthcare-10-00784-f002:**
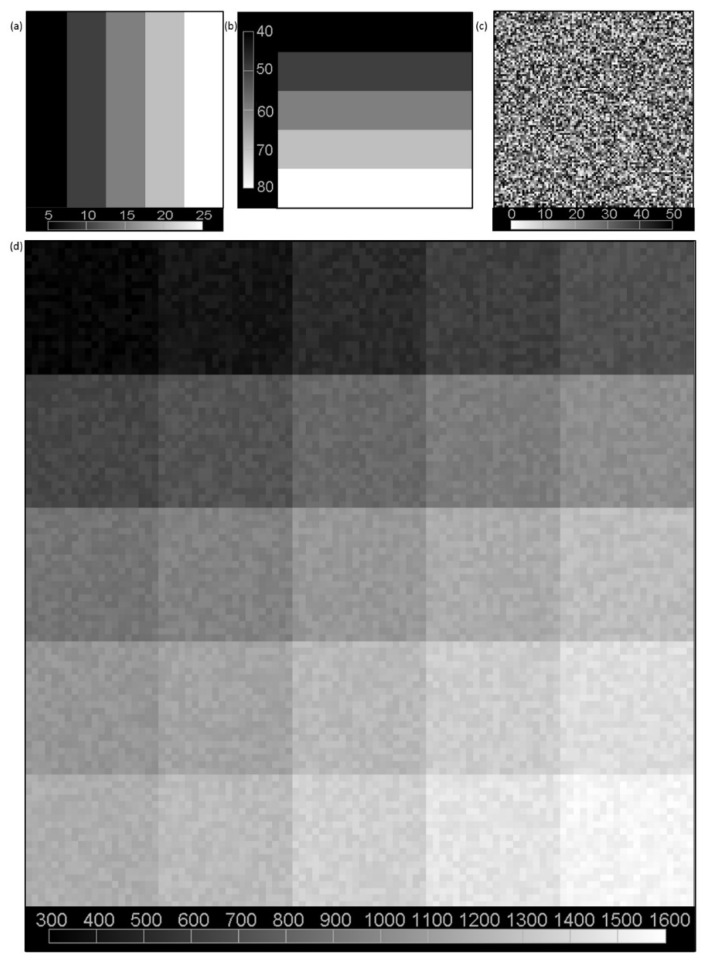
Simulated signal intensity with biexponential decay for TE = 9 ms. The signal intensity was created from two constant-amplitude components 1200 and 1800: (**a**) short T2 component with values of 5, 10, 15, 20, and 25 ms; (**b**) long T2 component with values of 40, 50, 60, 70, and 80 ms; (**c**) noise (level was set to 50); and (**d**) the simulated image consisting of 25 subimages (20 × 20 pixels).

**Figure 3 healthcare-10-00784-f003:**
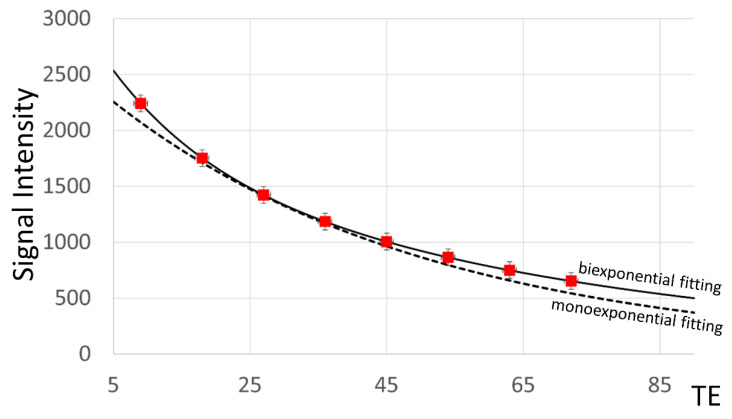
Signal-to-TE dependency for MRI examinations of the average signal of a single study from an automatically segmented AT region, proving that the signal is sufficient for biexponential fitting (solid line). In comparison, a monoexponential fitting result (dashed line) is presented, and it displayed a larger error of fit. Data points are marked in red.

**Figure 4 healthcare-10-00784-f004:**
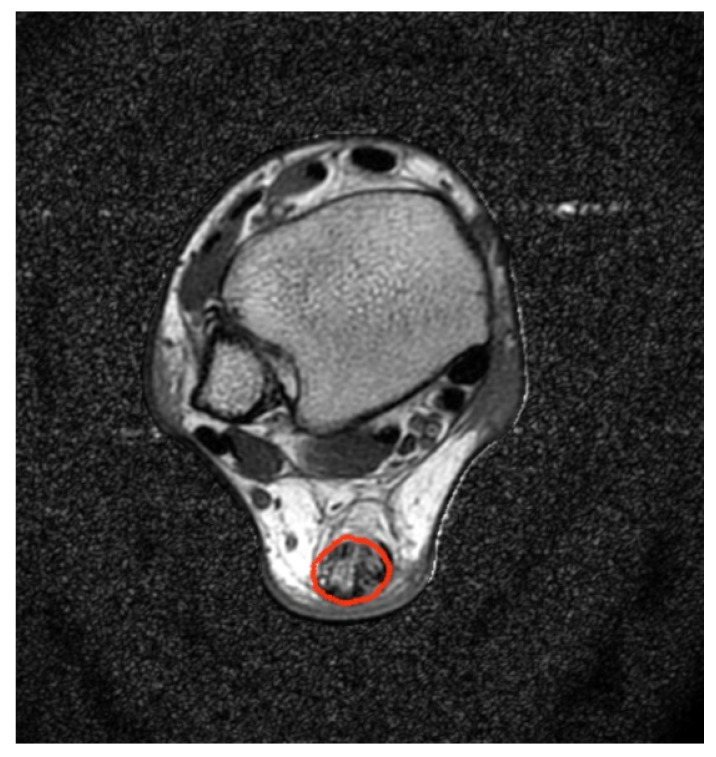
Intensity signal obtained with TE = 18 ms. The high level of noise in the surrounding air was noticeable. Segmentation of the AT region (outlined in red) was provided automatically. Biexponential T2 map reconstructions were performed based on segmented AT regions.

**Figure 5 healthcare-10-00784-f005:**
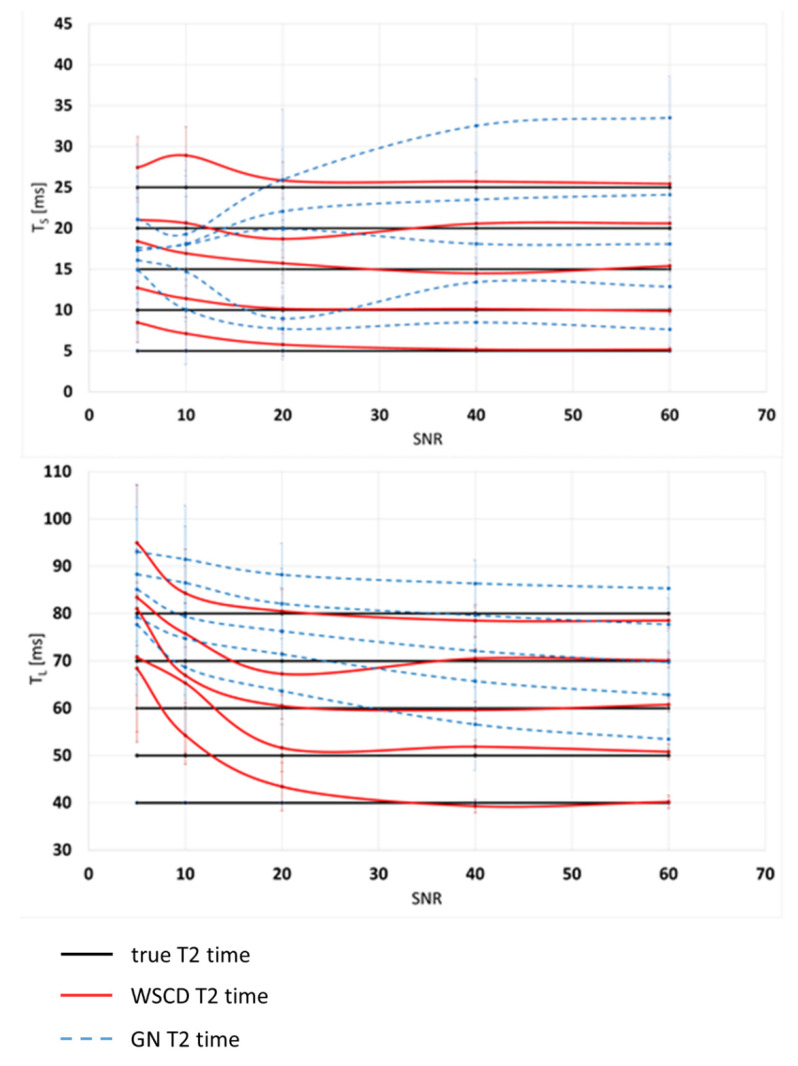
Accuracy of long (*T_L_*) and short (*T_S_*) T2 components from simulated data with the WSCD and GN methods. Colors: black, red, and blue denote the true T2, WSCD-reconstructed, and GN-reconstructed values, respectively. Error bars represent the SD. The reconstructed T2 values should be as close as possible to the true *T_S_* values of 5, 10, 15, 20, and 25 ms and true *T_L_* values of 40, 50, 60, 70, and 80 ms.

**Figure 6 healthcare-10-00784-f006:**
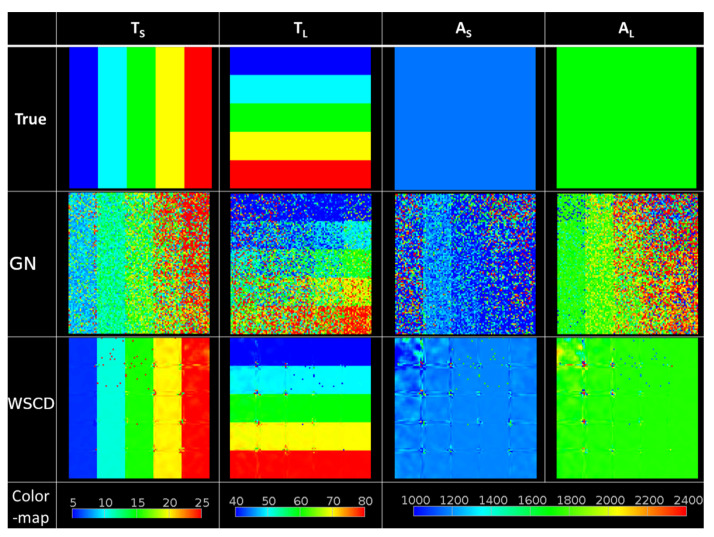
Comparison of biexponential reconstructions based on the GN and WSCD methods with the true *T_S_*, *T_L_*, A_S_, and A_L_ maps. True maps were used to create simulated intensity signals from which the biexponential reconstructions were obtained. The noise level was set to 100 and was normally distributed. The reconstruction maps were affected by noise; however, the WSCD method proved to be more accurate.

**Figure 7 healthcare-10-00784-f007:**
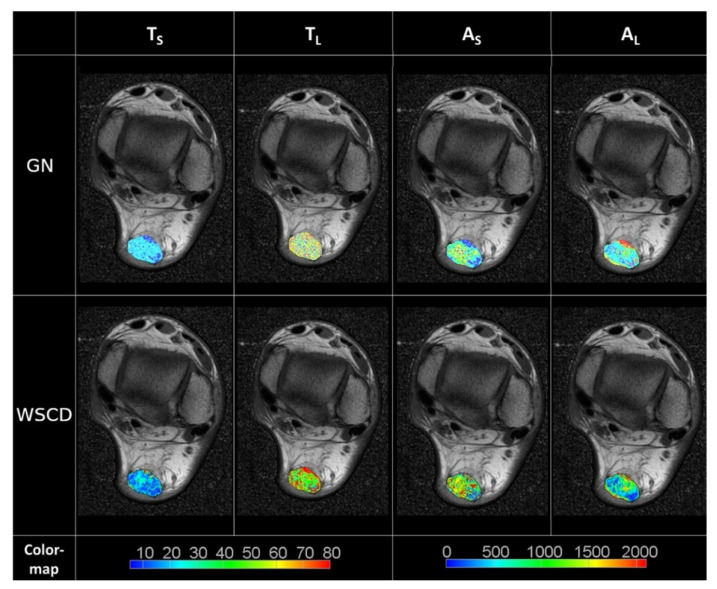
Comparison of biexponential reconstruction methods based on segmented AT regions in MRIs. The segmented and reconstructed AT regions were imposed on the initial MRI.

**Figure 8 healthcare-10-00784-f008:**
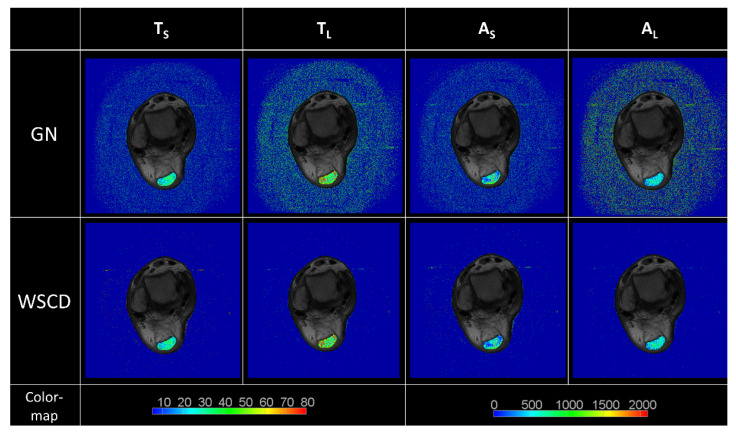
Signal-to-noise comparison between the WSCD and GN methods based on reconstructed T2 maps. The signal reference value was the average of the T2 time in the segmented AT region. The noise reference value was the mean of the T2 time in the air background. The noise level in the T2 maps reconstructed with the GN method was higher than that in the maps reconstructed with the WSCD method.

**Table 1 healthcare-10-00784-t001:** Detailed results of the accuracy of the GN and WSCD reconstruction methods. Significant deviations from true T_2_ time values for short (*T_S_*) and long (*T_L_*) components were obtained for all SNRs based on the GN method and for SNRs < 20 based on the WSCD method. The WSCD method with an SNR ≥ 20 did not show a significant deviation from the true T2 values. The statistical analysis was performed with the Wilcoxon signed-rank test with continuity correction. The numerical values of the mean difference from true T2 values, standard deviation from the pseudomedian, and confidence intervals (CI) of *T_S_* and *T_L_* are given in milliseconds.

	SNR	Method	Mean Difference	Standard Deviation	Wilcoxon Signed-Rank Test with Continuity Correction			
					Pseudo median	CI lower	CI upper	*p*
*T_S_*	5	GN	−1.14	17.06	−1.38	−1.82	−0.93	0.0000
		WSCD	3.41	5.73	1.30	0.41	2.18	0.0000
	10	GN	0.30	15.42	−0.51	−0.92	−0.11	0.0000
		WSCD	−2.00	5.05	−0.54	−0.81	−0.26	0.0000
	20	GN	2.23	11.13	1.66	1.25	2.40	0.0000
		WSCD	−0.51	5.73	−0.89	−2.18	0.41	0.0895
	40	GN	2.92	7.98	2.50	2.01	2.95	0.0000
		WSCD	−0.42	4.04	−0.05	−0.43	0.33	0.1315
	60	GN	2.97	6.76	2.50	2.30	2.71	0.0000
		WSCD	0.54	3.60	0.02	−0.03	0.07	0.2112
*T_L_*	5	GN	20.27	33.32	14.95	10.10	40.00	0.0000
		WSCD	14.05	10.46	14.05	6.65	21.44	0.0000
	10	GN	13.26	25.28	15.00	12.49	17.50	0.0000
		WSCD	6.36	12.89	4.90	−0.56	10.44	0.0017
	20	GN	13.44	22.06	12.51	12.50	14.99	0.0000
		WSCD	−0.50	6.43	−0.40	−1.73	0.94	0.3139
	40	GN	12.07	15.94	10.00	10.00	12.50	0.0000
		WSCD	−0.81	4.51	−0.20	−0.52	0.12	0.1025
	60	GN	8.80	13.96	7.50	7.50	10.00	0.0000
		WSCD	0.27	2.88	0.10	−0.16	0.40	0.2137

## Data Availability

The data that support the findings of this study are available from the corresponding author upon reasonable request.

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
