# Peer review of "A Weighted Stochastic Conjugate Direction Algorithm for Quantitative Magnetic Resonance Images—A Pattern in Ruptured Achilles Tendon T2-Mapping Assessment"

_healthcare, 2022, doi:10.3390/healthcare10050784_

Round 1
Reviewer 1 Report
In this article, the authors attempt to improve the T2-map using the weighted stochastic conjugate direction method (WSCD). Compared with the Gauss-Newton (GN) numerical optimization method used in the past, the WSCD method has higher anti-noise image reconstruction performance, and can distinguish two T2 components by this method. In addition, the problem of the elongation of the examination and long reconstruction times can be solved with this method. Finally, experimental results on simulated data and AT rupture MRIs demonstrate the superiority of WSCD over GN methods. However, there seems to be a lack of more effective explanations of real MRI data in the conclusion part, and it is recommended to further process and improve the data. The clinical application of this method is expected to provide surgeons with faster and more effective information during the pre-diagnosis process, thereby further exerting the role of MRI.
Here are some suggestions for the content:
- When introducing the calculation method of the signal in lines 74-78, the cited paper is missing, and the relevant citation of formula (1) should be added.
- The same problem as above occurs in formula (2).
- The description of lines 88-91 can be explained by adding a simple schematic diagram, which is more conducive to the reader's understanding.
- The descriptions of yP(t) and yQ(t) in formula (4) can be adjusted to the beginning of line 98, which is helpful for the coherence of the text. But now it is placed in line 101, which is far away from formula (4) and formula (5), which makes people confused.
- The MSE algorithm in line 99 is only mentioned in the abstract part, and the full name of the MSE should be declared here.
- There is a general lack of citations and arguments in the formula explanation stage and should be supplemented.
- In line 127, "no significant improvement is obtained" needs to be clearly quantified,what variable is less than how much is called no significant improvement, and what are these based on.
- The writing format of 8:17 minutes on line 153 needs to be confirmed to be correct.
- The writing format of the data following the in-slice resolution in line 154 is wrong.
- The legend of a) in Figure 1 increases from 6-24 interval 2, but the content in the textis from 5-25 interval 5 increments, the legend should be revised. The same problem occurs in b) c) d), in addition, the legend of b) will be more clear if it is placed vertically.
- The meaning of the abscissa and ordinate in Figure 2 should be indicated in the figure, and the meaning of the red squares in the figure is not explained. Such pictures should also add corresponding legends to facilitate readers' understanding, rather than just describing them in words.
- The description of Figure 4 and Figure 5 is not convincing. In the simulation experiment, it is obvious that the anti-noise ability of WSCD is stronger, but in real data, the SNR of GN method is obviously better than that of WSCD, which is in stark contrast to the results in the simulation experiments, this phenomenon should be explained more fully, and the SNR results of the two methods should also be presented in the results of the simulation experiments and compared at the same time.
- The SNR data of WSCD and GN methods in real MRI are not provided in this paper, and the results are described in a general way of size comparison, which is not convincing.
- The content of table1 is not very clear. The SNR in the table changes from 5-60, but the previous article does not seem to mention how to set the SNR or the reason for this setting.
- Figure 6 is missing a legend.
- The real MRI image only introduces one case, but it is stated in the data description section that there are 58 subjects. More real MRI data should be added and the results will be counted to make the conclusion more convincing.
Author Response
In this artcle, the authors attempt to improve the T2-map using the weighted stochastic conjugate direction method (WSCD). Compared with the Gauss-Newton (GN) numerical optimization method used in the past, the WSCD method has higher anti-noise image reconstruction performance, and can distinguish two T2 components by this method. In addition, the problem of the elongation of the examination and long reconstruction times can be solved with this method. Finally, experimental results on simulated data and AT rupture MRIs demonstrate the superiority of WSCD over GN methods. However, there seems to be a lack of more effective explanations of real MRI data in the conclusion part, and it is recommended to further process and improve the data. The clinical application of this method is expected to provide surgeons with faster and more effective information during the pre-diagnosis process, thereby further exerting the role of MRI.
Here are some suggestions for the content:
1. When introducing the calculation method of the signal in lines 74-78, the cited paper is missing, and the relevant citation of formula (1) should be added.
Response 1: According to the Reviewer’s suggestion the reference was added.
2.The same problem as above occurs in formula (2).
Response 2: According to the Reviewer’s suggestion the reference was added.
3. The description of lines 88-91 can be explained by adding a simple schematic diagram, which is more conducive to the reader's understanding.
Response 3: The schematic figure was added.
4. The descriptions of yP(t) and yQ(t) in formula (4) can be adjusted to the beginning of line 98, which is helpful for the coherence of the text. But now it is placed in line 101, which is far away from formula (4) and formula (5), which makes people confused.
Response 4: According to the Reviewer’s suggestion the descriptions were changed.
5. The MSE algorithm in line 99 is only mentioned in the abstract part, and the full name of the MSE should be declared here.
Response 5: The abbreviation was declared according to the Reviewer’s suggestion.
6. There is a general lack of citations and arguments in the formula explanation stage and should be supplemented.
Response 6: The citations and arguments were supplemented.
7. In line 127, "no significant improvement is obtained" needs to be clearly quantified,what variable is less than how much is called no significant improvement, and what are these based on.
Response 7: According to the Reviewer’s suggestion the explanation was added to the manuscript.
8. The writing format of 8:17 minutes on line 153 needs to be confirmed to be correct.
Response 8: We would like to confirm, that the time was 8 minutes and 17 seconds.
9. The writing format of the data following the in-slice resolution in line 154 is wrong.
Response 9: The format was changed.
10. The legend of a) in Figure 1 increases from 6-24 interval 2, but the content in the textis from 5-25 interval 5 increments, the legend should be revised. The same problem occurs in b) c) d), in addition, the legend of b) will be more clear if it is placed vertically.
Response 10: The legends were changed according to the Reviewer’s suggestion.
11. The meaning of the abscissa and ordinate in Figure 2 should be indicated in the figure, and the meaning of the red squares in the figure is not explained. Such pictures should also add corresponding legends to facilitate readers' understanding, rather than just describing them in words.
Response 11: The image was changed according to the Reviewer’s suggestion.
12. The description of Figure 4 and Figure 5 is not convincing. In the simulation experiment, it is obvious that the anti-noise ability of WSCD is stronger, but in real data, the SNR of GN method is obviously better than that of WSCD, which is in stark contrast to the results in the simulation experiments, this phenomenon should be explained more fully, and the SNR results of the two methods should also be presented in the results of the simulation experiments and compared at the same time.
Response 12: In fact, the reconstructed area in the “real” MRI data was the Achilles tendon region. In figure 5, the segmented and reconstructed Achilles tendon was imposed on the initial, non-reconstructed MRI. Therefore, the background should not be compared. The noise level was reduced in the Achilles tendon region. The image was changed, to emphasize the difference in Achilles tendon, and not in the background.
13. The SNR data of WSCD and GN methods in real MRI are not provided in this paper, and the results are described in a general way of size comparison, which is not convincing.
Response 13: The signal to noise ratios in the resultant T2 maps were added. The SNR in the WSCD method proved to be significantly lower than SNR in the GN method. We updated the manuscript by adding:
“To compare the accuracy of both methods based on real MRI data, the resultant SNR was calculated using the background noise in air as a reference. The signal was the aver-age value from the segmented AT T2 map. Noise was defined as the mean value in the background. The SNR values were compared with the WSRT results. A statistical analysis was performed with RStudio.
The mean value of the SNR for the resultant reconstructions was 25.51 for the WSCD method and 3.02 for the GN method, verifying the outstanding accuracy of the WSCD ap-proach. The WSRT statistical analysis revealed a significant deviation of the GN model from the WSCD model (p<0.0001).”
14. The content of table1 is not very clear. The SNR in the table changes from 5-60, but the previous article does not seem to mention how to set the SNR or the reason for this setting.
Response 14: According to the Reviewer’s suggestion in the methodology section the description of SNR analysis was extended by adding: “The arbitrary selection of noise levels resulted in constant SNR values throughout the im-age. The SNR in the simulated data was defined as the quotient of the signal value and noise value. Therefore, the images had SNR values equal to 60, 40, 20, 10, and 5. The re-sults were analysed according to these SNR values..”
15. Figure 6 is missing a legend.
Response 15: The legend was added.
16. The real MRI image only introduces one case, but it is stated in the data description section that there are 58 subjects. More real MRI data should be added and the results will be counted to make the conclusion more convincing.
Response 16: According to the Reviewer’s suggestion another figure was added, showing the difference in the SNRs between both methods.
Reviewer 2 Report
Explain all figures clearly and also cite them with proper references.
There are many grammatical and typo mistakes such as where nTE – is the number of TEs, αP,Q is the weightin
In Eq. (4) why variance is used as denominator.
Justify why it is assumed that “point Q belongs to the neighbourhood U and is not equal to the point P”
Can you justify Eq. 5 with proper details or methods?
Derive and explain Eq. (6). I am unable to understand it with present description.
Why ?0(0.75?, 1.25?, ?/2, ?/2, 0} in Eq. (7) similary for Eq. (8)
Eq. (9) is confusing. Unable to fetch the main idea.
More comparative analyses with statistical description are required.
Remove reference from conclusion section.
Author Response
Explain all figures clearly and also cite them with proper references.
Response 1: All figures are referred in the text.
There are many grammatical and typo mistakes such as where nTE – is the number of TEs, αP,Q is the weightin
Response 2: Grammar and spelling check was performed by American Journal Experts (aje.com).
In Eq. (4) why variance is used as denominator.
Response 3: According to the Reviewer’s suggestion the explanation was added:
“Weights are introduced to decrease the influence of signal noise. The local noise measure is assumed to be a variance (σ2) calculated for each voxel and each echo time within a window of a given radius. Therefore, to avoid overestimation of the noise within windows, the variance is used in the denominator of the abovementioned equation. The normalization factor is used to reduce the impact of noise on the considerably different signal values at points P and Q. A similar approach is used in anisotropic denoising algorithms [23,24].”
Justify why it is assumed that “point Q belongs to the neighbourhood U and is not equal to the point P”
Response 4: Point P is the window center point. Point Q is a point around the point P. When point P is equal to Q then both the weighting factor and the normalizing factor would be equal to 1. Therefore, it would have no impact on the total weight W(t).
Can you justify Eq. 5 with proper details or methods?
Response 4: The normalizing factor in the Eq. 5. is used to reduce the impact of noise on the considerably different signal values at point P and Q. The method is similar to the anisotropic denoising algorithm justified in the following references:
- Regulski P, Zieliński J, Borucki B, Nowiński K. Assessment of anisotropic denoiser enhanced cone beam CT for patient dose reduction International Journal of Computer Assisted Radiology and Surgery CARS suppl. 1, 10(1), 2015, pp. 295-296
- Zielinski, K. Nowinski, ‘‘Multi-step anisotropic denoiser scheme applied for cardiac non-contrast CT images’’, Int. J. of CARS, Vol. 9 (suppl. 1), 2014
Derive and explain Eq. (6). I am unable to understand it with present description.
Response 5: Equation was explained according to the Reviewer’s suggestion.
Why ?0(0.75?, 1.25?, ?/2, ?/2, 0} in Eq. (7) similary for Eq. (8)
Response 6: The issue was associated with incorrect formatting of the text. Now the formatting was corrected.
Eq. (9) is confusing. Unable to fetch the main idea.
Response 6: The equation was replaced with the following description of the idea:
“The segment |B0Bi1| is divided into m=20 subsegments that are equal in length. Let us consider the set of points (Ci) containing points B0 and Bi1 and the set of points Bik for which the value of the function g(Bik/m) is larger than the function value at two adjacent points g(Bi(k-1)/m) and g(Bi(k+1)/m); k ∈ (0,m).
More comparative analyses with statistical description are required.
Response 7: Comparative analysis using SNR between WSCD and GN method was added in the materials and methods section and in the results section:
“To compare the accuracy of both methods based on real MRI data, the resultant SNR was calculated using the background noise in air as a reference. The signal was the aver-age value from the segmented AT T2 map. Noise was defined as the mean value in the background. The SNR values were compared with the WSRT results. A statistical analysis was performed with RStudio.
The mean value of the SNR for the resultant reconstructions was 25.51 for the WSCD method and 3.02 for the GN method, verifying the outstanding accuracy of the WSCD ap-proach. The WSRT statistical analysis revealed a significant deviation of the GN model from the WSCD model (p<0.0001).”
Remove reference from conclusion section.
Response 8: According to the Reviewer’s suggestion the reference was removed.
Reviewer 3 Report
Dear authors,
The topic is very interesting, however I have some suggestions:
- Introduction: maybe you can insert one paragraph about achilles tendon and its roture
- Results: you need to describe all the sequence of the results, beginning with the number of images etc
- discussion: you must compare with literature your results and use references
best regards
Author Response
The topic is very interesting, however I have some suggestions:
- Introduction: maybe you can insert one paragraph about achilles tendon and its roture
Response 1: According to the Reviewer’s suggestion the following paragraph was added:
“The incidence of AT rupture is high – approximately 7-18 per 100,000 in the general population – and it may occur during spontaneous recreational activity or during profes-sional sports activities [17,18]. Compared to that for other tendons and ligaments, the AT healing process is prolonged due to comparatively poor blood supply [19,20]. Major changes in the AT mainly occur during the first half year after injury, resulting in changes in the MRI of the tendon [21].”
- Results: you need to describe all the sequence of the results, beginning with the number of images etc
Response 2: According to the Reviewer’s suggestion the number of total images (slices of MRI) were added. The results section have been rearranged to reflect the sequence in the method section.
- discussion: you must compare with literature your results and use references
Response 3: Discussion was extended according to the Reviewer’s suggestion. The following paragraph was updated:
“Attempts have been made to solve the abovementioned problems with monoexpo-nential models. Improvements in the accuracy and precision of low-SNR T2 maps were obtained by Raya et al. [10], who developed two noise-corrected fitting methods: fitting to a noise-corrected exponential and fitting the noise-corrected squared signal intensity to an exponential. Sandino et al. [7] introduced a pixelwise nonlinear regression method by us-ing SNR-scaled image reconstruction and truncating low-SNR measurements. Akcakaya et al. [31] developed an improved T2-based, balanced steady-state free-precession se-quence and a signal relaxation curve fitting method. To the best of our knowledge, a few quality-improving methods for biexponential models have also been presented, including methods based on optimized TE sampling procedures [27,32], nonselective radio fre-quency pulses [12] and separate nonlinear filters [33–35]. Huang presented a study in which a nonlinear mixed-effect model was used for the reconstruction of biexponential time maps, and it improved the accuracy of parameter estimation [36]. Shao et al. [37] used a maximum likelihood estimation algorithm for noise estimation in a biexponential approach. However, none of these methods considered a postacquisition optimization or assessment of the reconstruction model. The clinical aspect of this research has been discussed by Kapinski et al. [38], who monitored the healing of postrupture ATs. The clinical application of this method is ex-pected to provide physicians and surgeons with faster and more effective information during the prediagnosis process. Furthermore, the presented biexponential reconstruction was used in assessments of the anatomical structure of the temporomandibular joint [39].”
Round 2
Reviewer 2 Report
The authors have improved the paper. So, I tend to accept it its present form.